# American crows that excel at tool use activate neural circuits distinct from less talented individuals

LomaJohn T. Pendergraft [1,2] ✉, John M. Marzluff [1], Donna J. Cross [3], Toru Shimizu[4] & Christopher N. Templeton [5]

Tools enable animals to exploit and command new resources. However, the neural circuits underpinning tool use and how neural activity varies with an animal's tool proficiency, are only known for humans and some other primates. We use 18F-fluorodeoxyglucose positron emission tomography to image the brain activity of naïve vs trained American crows (*Corvus brachyrhynchos*) when presented with a task requiring the use of stone tools. As in humans, talent affects the neural circuits activated by crows as they prepare to execute the task. Naïve and less proficient crows use neural circuits associated with sensory- and higher-order processing centers (the mesopallium and nidopallium), while highly proficient individuals increase activity in circuits associated with motor learning and tactile control (hippocampus, tegmentum, nucleus basorostralis, and cerebellum). Greater proficiency is found primarily in adult female crows and may reflect their need to use more cognitively complex strategies, like tool use, to obtain food.

Tool use was once considered unique to humans, yet is now known from multiple vertebrates, and even among some invertebrates[1]. Despite its occurrence across a diversity of taxonomic classes, tool use behavior is not universal or even common among other taxa within the same family or genus, even though most possess the requisite morphology, as evidenced by their capacity to learn tool use in captivity[2–4]. Considering the advantages it affords the practitioner, why don't we see more widespread tool use among the other species belonging to groups where at least one species utilizes tools?

While scientists no longer consider advanced cognition and behavioral flexibility to be requisite for tool use – as demonstrated by genetically heritable and stereotyped tool use behaviors found among arthropods[5] and fish[6] – these traits tend to be common among species that must learn to use tools[1]. Researchers may be able to explore why more animals don't use tools by examining the brain's active regions when the subject uses a tool. Two distinct neural circuits guide and shape tool use in humans and non-human primates; one circuit is responsible for the semantic knowledge of the tool while the other

controls the learned motor skills required to effectively use the tool[7–9]. However, whether other tool-using animals possess functionally similar neural circuitry is not known.

Many members of the avian family *Corvidae* (corvids) show complex behavior comparable to the great apes; both groups possess equivalent forebrain neuronal counts[10] and can solve novel challenges with similar speed and flexibility[11]. However, only the New Caledonian crow (*Corvus moneduloides*) is a habitual tool user[12]; other corvids only occasionally use tools in the wild[13], though they are capable of learning a variety of tools in captivity[14]. Individual variation in the ability of occasional tool users (such as American crows; *Corvus brachyrhynchos*) to learn and master the use of novel tools offers a window into the physical and neural attributes that favor adopting tool use.

Here, we capture 16 wild American crows and use 18F-fluorodeoxyglucose positron emission tomography (FDG-PET) to explore the neurological activity associated with their proficiency at learning the Aesop's fable paradigm (hereafter Aesop's fable task) requiring the use of a novel tool (Fig. 1). This task requires the subject

[1]University of Washington, School of Environmental and Forest Sciences, Seattle, WA, USA. [2]University of Washington, Department of Psychology, Seattle, WA, USA. [3]University of Utah, Department of Radiology and Imaging Sciences, Salt Lake City, UT, USA. [4]University of South Florida, Department of Psychology, College of Arts & Sciences, Tampa, FL, USA. [5]Western Washington University, Biology Department, Bellingham, WA, USA. ✉e-mail: pendel@uw.edu

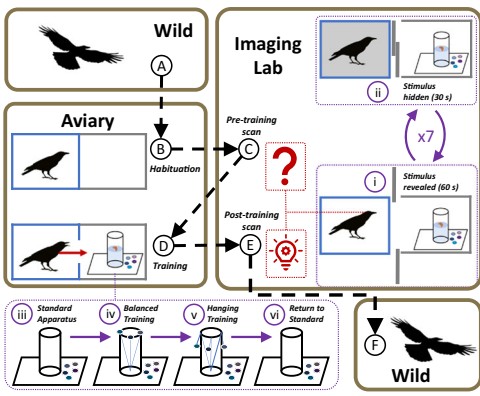

**Fig. 1 | Overview of our methodology.** We captured crows from the wild (**A**) and housed them in an outdoor aviary. For two weeks, we regularly habituated the crows to the conditions of the imaging process, including temporarily placing them in a smaller cage enclosed by panels that blocked all external view except that of an (empty) adjacent stimulus stage (**B**). During habituation, they could not enter the empty stimulus stage. We transported crows from the aviary to the imaging lab for their baseline pre-training scan; this was the first time the naïve crows were exposed to the materials they would use to perform an Aesop's fable task (**C**). During the imaging process, we stimulated the crows by alternatively revealing (**i**) and hiding (**ii**) the interior of the well-lit stimulus stage and the task materials within. We returned crows to the aviary and modified the habituation process to train them to the task; they could now access the materials within the stimulus stage (**D**). We used a standard apparatus (**iii**) during training until we had established the maximum reachable depth for the crow – if we lowered the water level beyond this point, the crow could not reach the food without dropping at least one stone into the tube. We taught crows that the stones could be used as tools to raise the water level by introducing training stones to the rim of the tube (**iv**). Training stones are connected to the tube's interior via lines, preventing them from falling outside the tube, and crows usually push them in as they reach for the food. We increased the difficulty by suspending the training stones along the tube's exterior, which prevented them from being knocked inside accidentally (**v**), before finally removing them and returning to the standard arrangement (**vi**). After imparting differential task proficiency, we returned the crows to the imaging laboratory for their post-training scan (**E**). We released crows back to the wild several days after the final scan (**F**).

to drop stones into water to raise its level and obtain an out-of-reach food item and has been used to test the cognitive capabilities of multiple corvid species[15]. We have previously utilized FDG-PET imaging to examine avian neural activity[16–19] because, unlike many in vivo imaging modalities, the subject can be awake and unrestrained when exposed to the experimental stimulus, thus reducing a source of stress that can confound the recorded neurological activity. We scanned all crows twice; first before they had ever been exposed to the Aesop task (hereafter Pre-training scan), and second after the birds were trained to retrieve food from the Aesop tube (hereafter Post-training scan). Thus, the imaging procedure and stimulus remained precisely the same between pre- and post-training scans, with the only difference being the crows' level of experience using tools to solve the visible task. We expect proficient crows to show increased activity (compared to naïve or less proficient crows) in brain regions previously identified as enlarged in New Caledonian crows that regularly use tools: the mesopallium, striatal complex, septum, and tegmentum[20]. We also expect proficient birds to activate their NCL due to its association with tool use[21], and hippocampus due to its association with memory[22]. We find activity within the pallial regions (mesopallium and nidopallium) for naïve and less proficient crows, while highly proficient crows show activation in memory, motor, and tactile associated regions (hippocampus, tegmentum, nucleus basorostralis, and cerebellum). We additionally use behavioral observation and computerized tomography (CT) to examine the physical factors (such as sex, body size, and brain volume) associated with task proficiency. We expect hunger

level or body condition[23], age or experience[24], sex or social status[25], and temperament[26] to affect motivation to use and master tools, and find that age and sex are the greatest predictors of tool mastery (adult females are most proficient).

## Results

### Task proficiency
Crows varied dramatically in their degree of mastery of the Aesop's fable task, with only four of the 16 crows fully mastering it (Fig. 1vi; high-proficiency). Of the 12 birds that did not master the task, four never dropped any balanced training stones (Fig. 1iv; low-proficiency) and eight never picked up + dropped any hanging training stones (Fig. 1v; medium-proficiency).

### Pre- and Post-training differential brain activity
Compared to their post-training scan, the crows showed three distinct regions of significantly higher FDG uptake within the brain during their pre-training scan, regardless of their eventual proficiency with the task. Starting with the most rostral region (Fig. 2A), the crows showed bilateral subthreshold activation within the lateral mesopallium, with peak loci near the border between mesopallium and nidopallium (left hemisphere: 8.2% more activity, $Z = 4.53$; right hemisphere: 8.6% more, $Z = 4.22$). The second region (Fig. 2B) contained the most widespread subthreshold activation and included loci in the right hemisphere dorsal mesopallium (7.4% more activity, $Z = 4.82$) and dorsal nidopallium (7.2% more, $Z = 5.24$). The latter continued caudally toward a more ventral portion of the nidopallium (6.9% more, $Z = 4.31$) of the right hemisphere. The presence of subthreshold activity (albeit less widespread) and a locus in the left hemisphere mesopallium (5.1% more activity, $Z = 4.27$) suggest a limited degree of bilaterality to this activity. The region of significantly increased activity in the caudal nidopallium (Fig. 2D) appeared to be lateral to the medial portion of the nidopallium caudolaterale (NCLm); while the activities were bilaterally observed (left hemisphere: 8.9% more activity, $Z = 4.01$; right hemisphere: 7.8% more activity, $Z = 4.57$), the subthreshold activity was more widespread in the right hemisphere. There was an additional region (Fig. 2C) of notably higher activity that did not exceed the Z-threshold of 4.0; it was also located within the caudal nidopallium, though more rostrally to the previous region and primarily limited to the left hemisphere (8.2% more, $Z = 3.80$). See Supplementary Fig. 1 in SI for differential activity patterns throughout the entire brain.

We found no differences in post-training neural activity that were consistent across all crows. However, when we subdivided the crows by task proficiency, we found that the high-proficiency birds showed significantly increased activity medially within their left hemisphere tegmentum after their training, relative to before (8.2% more, $Z = 4.16$, Fig. 3). This activity was found in a small region just caudal to the tuberal nucleus, possibly including the ventral tegmental area. See Supplementary Fig. 2 in SI for differential activity patterns throughout the entire brain.

### Proficiency-based differential brain activity
After training, crow proficiency at solving the Aesop's fable task was associated with differences in brain activity. Compared to the birds with low-proficiency, crows that were highly proficient at solving the task showed significantly more activity within the right hemisphere hippocampus (9.1% higher activity, $Z = 4.44$) and dorsally within both hemispheres of the caudal cerebellum (left: 9.8% more activity, $Z = 4.89$; right: 15% more activity, $Z = 4.23$), along with notably higher activity in the left hemisphere nucleus basorostralis (3.3% more activity, $Z = 3.77$) upon seeing the Aesop's task during their post-training scan (Fig. 4A). By contrast, low-proficiency crows showed notably more activity within the left hemisphere mesopallium, adjacent to (and possibly including) the mesopallium ventro-lateralis (MVL) (10.3% more activity, $Z = 3.92$), and the left hemisphere hyperpallium apicale

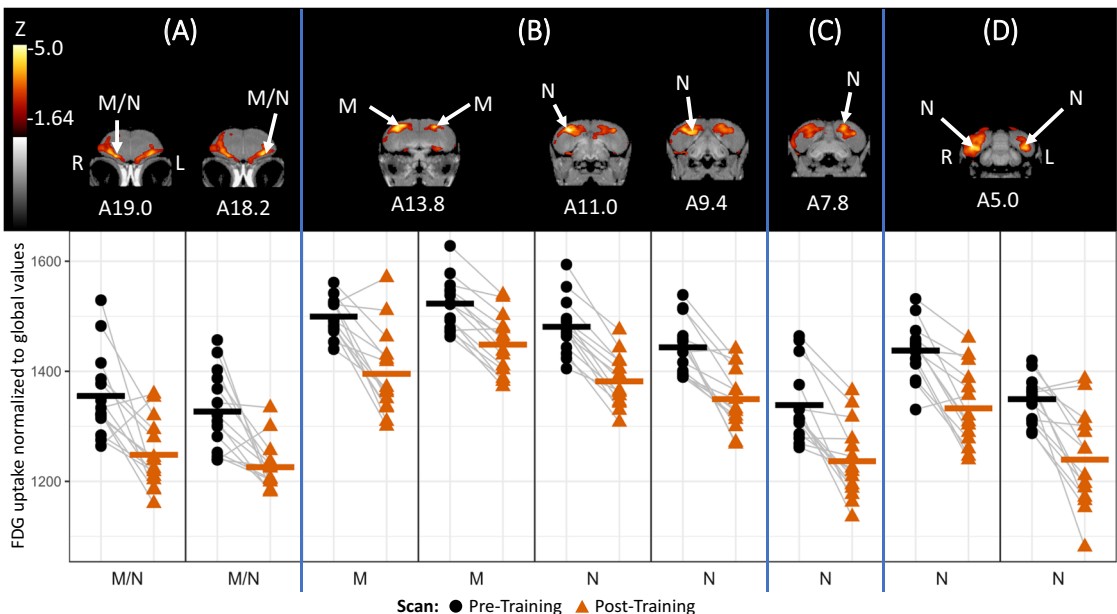

**Fig. 2 | Pre- versus post-training differential brain activity for all crows.** Top: Coronal view of voxel-wise subtractions (converted to Z-score maps) showing differential FDG uptake at the indicated region for all the crows' pre-training scan compared to their post-training scan (*n* = 14 crows). M mesopallium, N nidopallium. *Z*-score map is superimposed atop a composite (*n* = 4 scans) structural MRI of the American crow brain. Slices are arranged from most rostral to most caudal, and slice coordinates refer to an established jungle crow atlas[64]. Note that the atlases we used disagree on the loci caudal to A9.4; the jungle crow atlas[64] identifies them as nidopallium caudale, the carrion crow atlas[66] as nidopallium. Bottom: individual normalized (global) uptake values obtained from VOI's centered on peak activation coordinates. Gray lines link the pre-training (black circle) and post-training (dark orange triangle) scan values for each individual crow. Bold horizontal lines indicate group means for pre-training (black) and post-training (dark orange) scans. Vertical blue lines separate distinct regions of activity (**A**–**D**). Note that the activity at (**C**) within the nidopallium did not exceed the Z-threshold for statistical significance. Source data are provided as a source data file.

(21.6% more, *Z* = 3.79) compared to the high-proficiency birds (Fig. 4B), though neither region exceeded the *Z*-threshold for statistical significance (*Z* = 4.0). Although they showed much individual variability, the mean activity of medium-proficiency crows tended to be intermediate between that of the high- and low-proficiency birds in the regions previously mentioned (Fig. 4A, B). Additionally, the medium-proficiency crows had notably more activity laterally within the left hemisphere mesopallium (23.9% more activity, *Z* = 3.90, Fig. 4C) compared to the highly proficient birds. See Supplementary Figs. 3–5 in SI for differential brain activity throughout the entire brain for each comparison.

We additionally imaged the four highly proficient crows one final time after their post-training scan. During this 3rd scan, we allowed the bird to enter the stimulus stage and directly interact with the Aesop task apparatus. Brain activity during tool use differed little from post-training observation of the apparatus (see Supplementary Fig. 6 in SI for localized areas consistent with the nidopallium or arcopallium; one *Z* = 4.0, all others <3.4).

**Crow individual measures**

We sampled 10 males and 6 females, with the sexes equally divided by age (*n* = 3 adult and 3 subadult females, *n* = 5 adult and 5 subadult males). They varied in size (culmen length: mean ± SD = 50.0 ± 2.9 mm), body condition (0.00 ± 18.08 g), level of nervousness (31.94 ± 10.95 movements/min), and brain volume during their first scan (absolute brain volume: 7.56 ± 0.68 cm³; relative brain volume: 00 ± 0.50 cm³). Though not significant, there was a slight decrease in the crows' level of nervousness during their time in captivity (first measure: 38.00 ± 11.69 movements/min; second measure: 33.50 ± 13.17 movements/min; $t_{15}$ = 1.94, *P* = 0.07). Despite only having access to food for several hours each day, the crows gained a significant amount of weight during their time in captivity (capture: 387.5 ± 31.4 g; release: 415.0 ± 37.87 g; $t_{15}$ = 6.12, *P* < 0.001). See Supplementary Table 1 in SI

for the individual measures of each crow. As expected, many of the individual measures were correlated, which reflected typical allometry and sex differences (females are smaller; Supplementary Table 2 in SI). In addition, crows with high body condition scores were less nervous than those with low scores (*r* = −0.58, $t_{14}$ = −2.69, *P* = 0.017).

**Individual factors and task proficiency**

A crow's estimated age was the most important factor associated with proficiency; it was the only competitive model and garnered 90% of the weight of evidence, primarily because the high-proficiency group consisted exclusively of adults (Table 1, Fig. 5A). Despite this, age alone is an unreliable predictor of crow task proficiency because the adult crows were also the demographic with the worst performance; three of the four remaining adults never progressed beyond low-proficiency. By contrast, seven of the eight subadults achieved medium-proficiency with the task. This reveals a curious trend where adult crows are highly bimodal in tool adoption, whereas the subadults are consistently intermediate.

Within the adult crows, the models for absolute brain volume, sex, and body size were all competitive, collectively garnering 81% of the weight of evidence, although the model for sex failed to converge properly because all the adult females solved the task (Table 2). The successful adult crows were primarily female (Fig. 5C) and tended to have smaller culmen length (did solve: mean ± SD = 46.5 ± 3.0 mm; did not solve: 51.3 ± 0.96 mm) and smaller absolute brain volume than unsuccessful adults (did solve: 7.05 ± 0.45 cm³; did not solve: 8.00 ± 0.64 cm³) (Fig. 5B, D). Although not competitive, the models indicate the adults that mastered the task also had a slight tendency to be in better body condition (did solve: 8.39 ± 17.68 g; did not solve: −4.79 ± 3.67 g), less nervous (did solve: 25.5 ± 7.56 move/min; did not solve: 34.0 ± 11.75 move/min), and have less variable relative brain volume (did solve: −0.12 ± 0.33 cm³; did not solve: 0.08 ± 0.65 cm³) compared to the other adults (Supplementary Fig. 7 in SI).

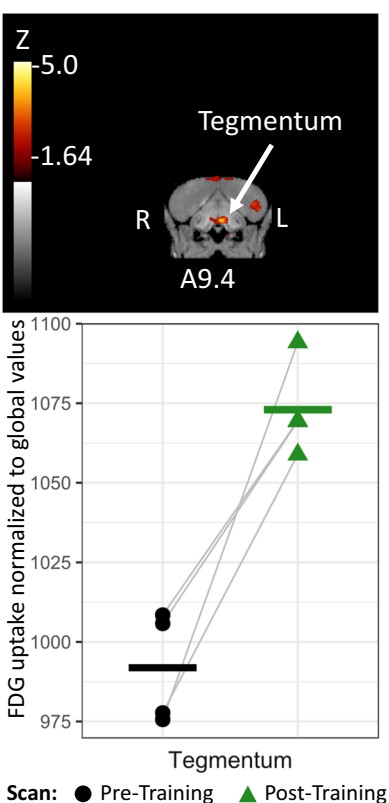

**Fig. 3 | Post- versus pre-training differential brain activity for highly proficient crows.** Top: Coronal view of voxel-wise subtractions (converted to Z-score map) showing differential FDG uptake at the indicated region for the high task proficiency crows' post-training scan compared to their pre-training scan ($n = 4$ crows). Z-score map is superimposed atop a composite ($n = 4$ scans) structural MRI of the American crow brain. Slice coordinates refer to an established jungle crow atlas[64]. Bottom: individual normalized (global) uptake values obtained from VOI's centered on peak activation coordinates. Gray lines link the pre-training (black circle) and post-training (green triangle) scan values for each individual crow. Bold horizontal lines indicate group means for pre-training (black) and post-training (green) scans. Source data are provided as a source data file.

Of the three within-adult competitive models, sex appears to be the main factor. The three adult females that solved the task represented 100% of that demographic in our sample group, whereas the single male in this proficiency group represented only 20%. By contrast, three of the four low-proficiency crows were adult males. The other two factors – body size and absolute brain volume – are likely only competitive because of the strong allometric correlation between sex, culmen length, and brain volume (Supplementary Table 2 in SI); female American crows are smaller than males[27] and smaller animals tend to have correspondingly smaller brains[28]. Relative brain volume corrects for this allometry, and it shows no relationship with task proficiency (Supplementary Fig. 7 in SI).

## Discussion

Imaging brain activity as individual birds were challenged to master the use of a novel tool provided a unique window into the neural changes that underpin learning invertebrates. Because the Aesop's fable task was difficult for crows to learn, we could separate birds into proficiency categories and relate their physical features and neural activity to their degree of task mastery. We were able to track approximate synaptic activity throughout the whole brain of each bird before and after they became familiar with the tool-using task because of the relatively non-invasive imaging technique we employed (PET).

Crows were first exposed to the Aesop's fable task during their pre-training scan, and they showed significantly increased FDG uptake, a surrogate of brain activity, throughout their mesopallium and nido-pallium compared to their post-training scan. This activity was nearly universal among all crows, regardless of their eventual task proficiency. The low- and medium-proficiency birds continued to show elevated activity in their mesopallium when scanned after weeks of experience with the task (compared to the high-proficiency birds), although the activity of the low-proficiency birds varied greatly between individuals compared to the medium- and high-proficiency groups. Both regions are known to be heterogenous entities in connection and function, and are involved with various sensory information and higher-order cognitive ability[29–31]. The mesopallium is an associative forebrain region that does not receive direct sensory information from the thalamus and has been suggested to be correlated with behavioral innovation and flexibility[32]. A sub-region within the nidopallium, the NCL in particular, has been compared to the mammalian prefrontal cortex in controlling executive function, working memory, planning, flexible thinking, and attending objects of interest, particularly if the object is associated with a reward[29,30,33]. However, much of our observed nidopallial activity occurred more rostrally (Fig. 2, Supplementary Fig. 1 in SI) in areas associated with processing trigeminal, visual, and auditory sensory information[34]. The nidopallium was comparably less active during a crow's second scan, thus this activity may have been caused by neophobic crows closely attending to the unfamiliar stimuli surrounding their first experience with the imaging process[35]. This observation contrasts with the elevated mesopallial activity among most of the less proficient crows during their second imaging experience. The mesopallial activity suggests that some of this activity was caused by the birds' cognitive processing associated with formulating a food retrieval strategy from the apparatus, rather than from neophobia.

After they learned to fully solve the Aesop's fable task, high-proficiency crows showed increased activity in their cerebellum, hippocampus, and nucleus basorostralis during their post-training scan compared to the low-proficiency crows. The hippocampus is strongly associated with memory[22], as is the cerebellum[36]. The cerebellum is additionally associated with motor learning, motor control, perception[37,38], and tool use[39]. Studies examining the cerebellum in pigeons suggest that the activity in the caudal cerebellum may be part of the oculomotor cerebellum[40]. With limited studies on the crow (or other passerine species) cerebellum, we cannot make further interpretations for this region. The nucleus basorostralis receives a trigeminal projection through the principal sensory trigeminal nucleus and is associated with processing somatosensory information from the beak[41,42]. High-proficiency crows also showed increased activity within their tegmentum after they mastered the task compared to their own brain activity when naïve. The tegmentum is a highly heterogenous region generally involved with motor control, though it also contains sensory nuclei and autonomous control centers[43]. The observed activity may include the ventral tegmental area (VTA), a region linked with the reward network and behavioral reinforcement[44]. If so, the reward pathway was selectively activated in the successful crows when they saw the apparatus.

The activity in the memory, motor, and tactile regions suggest that much like the mental prep of a ski racer entering the starting gate – visualizing a path through the course, the location of each bump and obstacle, the areas to speed up or slow down – the high-proficiency birds were preparing to enter the stimulus stage and solve the task. This is reflected in how the groups behaviorally interacted with the task in the weeks preceding their second imaging session. While the low-proficiency birds occasionally attempted to retrieve the food (without interacting with the stones), they usually gave up after a few attempts. They retreated to the training cage for the remainder of the training session. By contrast, the high-proficiency crows quickly,

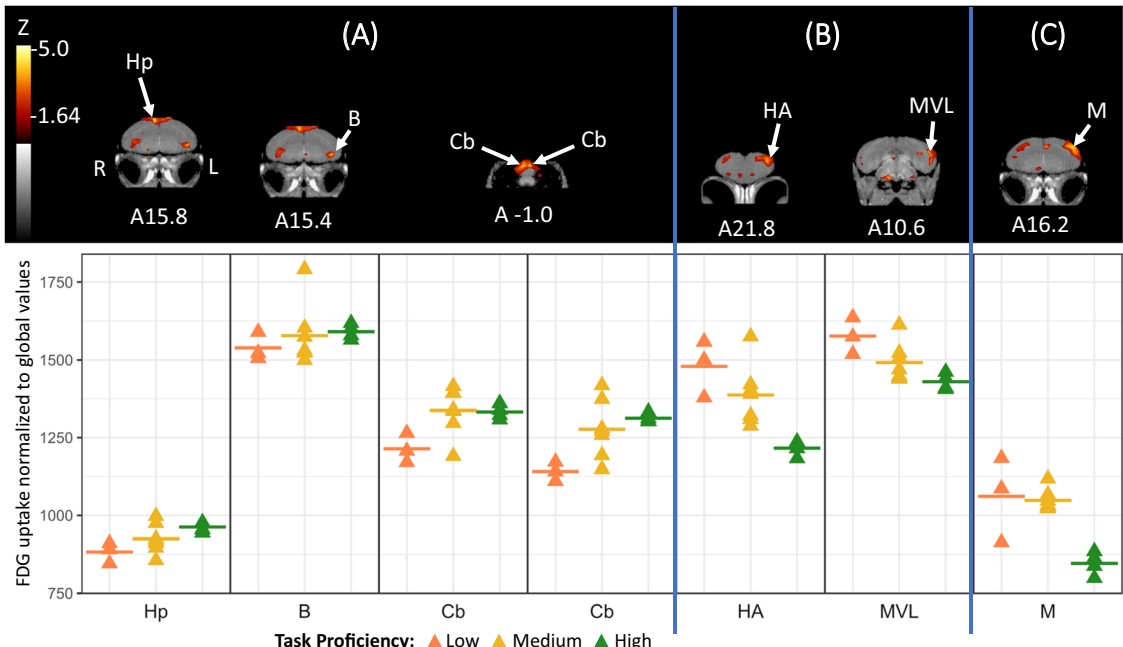

**Fig. 4 | Post-training differential brain activity for all proficiency groups.** Top: Coronal view of voxel-wise subtractions (converted to Z-score maps) showing differential FDG uptake by proficiency during the post-training scan. Vertical blue lines separate the high- vs low-proficiency (**A**, *n* = 7 crows), low- vs high-proficiency (**B**, *n* = 7 crows), and medium- vs high-proficiency (**C**, *n* = 11 crows) comparisons. Hp hippocampus, B nucleus basorostralis, Cb cerebellum, HA hyperpallium apicale, MVL mesopallium ventro-lateralis, M mesopallium. Z-score map is superimposed atop a composite (*n* = 4 scans) structural MRI of the American crow brain. Slices are arranged within each comparison from most rostral to most caudal, and slice coordinates refer to an established jungle crow atlas[64], although the peak activity within the cerebellum is more caudal than the atlas border (it is to scale). Bottom: individual normalized (global) uptake values from all crows (*n* = 14 crows) obtained from VOI's centered on peak activation coordinates. Horizontal lines indicate group means for birds in the low (orange), medium (yellow), and high (green) proficiency groups. Note that the nucleus basorostralis, hyperpallium apicale, mesopallium ventro-lateralis, and mesopallium did not exceed the Z-threshold for statistical significance. Source data are provided as a source data file.

consistently, and repeatedly picked up and dropped stones into the tube to bring the food within reach. This deliberate and repetitive action is highly conducive to memory formation and motor learning[45]. Unlike the consistent activity of the high- and low-proficiency groups, the medium-proficiency crows showed more individual variability in activity in the cerebellum, hippocampus, and nucleus basorostralis (Fig. 4A). This is likely because these birds varied in their interactions with the task; some medium-proficiency birds quickly gave up their attempts to retrieve the food after determining it was beyond their reach, while others persisted in repeatedly attempting to retrieve the out-of-reach food. This persistence was reinforced by the occasional success because we lowered the task difficulty if the crow failed to solve for three consecutive sessions.

These findings partially support our a priori hypotheses about which regions would be more active for the varying proficiency levels; while highly proficient birds did utilize their hippocampus and tegmentum compared to less proficient (or naïve) birds, the pallial regions were primarily active in naïve individuals as opposed to proficient crows. These results suggest that the high-proficiency crows no longer needed to devote higher-order cognitive power to the task because they already knew how to solve it. This change in relative neural activity as the crows progressed from naïve to varying degrees of tool proficiency resembles the differential neural activity observed between novice and elite human athletes[46,47]. Compared to more experienced athletes, novices have greater activity in the inferior frontal gyrus, superior frontal gyrus, and prefrontal cortex- regions that have been likened to the avian nidopallium[29]. However, as the athletes become more skilled, their neural activity shifts away from executive function to supplementary motor areas and the cerebellum. Evidently, highly proficient human athletes and crow tool users both rely more on learned

motor skills instead of executive planning to overcome familiar challenges in their respective areas of expertise.

Avian brains are highly lateralized[48], and we found differing levels of hemispherical bias in the active regions of this study, which are consistent with previous studies using American crows and PET imaging[17]. We found no difference in the crows' gaze direction (Supplementary Fig. 8 in SI), thus this hemispherical bias in activity cannot be explained from asymmetrical sensory input. Low-proficiency birds showed a left hemisphere bias in both hyperpallium apicale and MVL activity, consistent with previous research demonstrating that visual object processing is dominant in the left hemisphere[48,49]. Research on domestic chicks shows that the right hemisphere hippocampus is more sensitive to geometric spatial information[50]; our results could therefore indicate that the high-proficiency crows mentally recalled their direction of approach and position of the apparatus when they interacted with the Aesop's task during their training in the aviary. Pigeons with experience in long-distance navigation also show later-alization, with increased volume in their right hemisphere meso-pallium and nidopallium compared to pigeons without experience[51]; this bias in favor of the right hemisphere for cognitively demanding challenges supports our own findings. Finally, the right hemisphere has been extensively linked to novelty responses[35], thus the right hemisphere bias in the mesopallial/nidopallial activity of naïve birds may be due to unfamiliarity with the apparatus.

Tool use is evidently difficult for captive American crows to learn; task proficiency was not universal, some birds never progressed beyond low-proficiency, and the individuals that ultimately mastered the task required several weeks of training to learn to associate dropping stones into the tube's interior with raising the water level (see learning curves in Supplementary Fig. 9 in SI). Yet, despite the difficulty, some crows successfully overcame this challenge and

**Table 1 | Individual variable model selection (multinomial) for a crow's likelihood of progressing beyond low task proficiency**

| Model | AICc | Δ AICc | Wi | Intercept ± SE | Coefficient ± SE |
|---|---|---|---|---|---|
| **Age[†]** | **29.62** | **0.00** | **0.90** | **Med: −1.10 ± 1.15** <br> **High: 0.29 ± 0.76** | **Med: 3.04 ± 1.57** <br> **High: −8.87 ± 73.18** |
| Size | 37.13 | 7.52 | 0.02 | Med: 0.21 ± 12.17 <br> High: 21.62 ± 14.60 | Med: 9.61e-03 ± 0.24 <br> High: −4.50e-01 ± 0.30 |
| Absolute brain volume | 37.27 | 7.65 | 0.02 | Med: 3.04 ± 7.56 <br> High: 18.11 ± 11.52 | Med: −0.30 ± 0.97 <br> High: −2.45 ± 1.57 |
| Null | 37.27 | 7.65 | 0.02 | Med: 6.31e-01 ± 0.61 <br> High: −3.38e-06 ± 0.71 | |
| Sex[†] | 38.10 | 8.48 | 0.01 | Med: 0.96 ± 1.22 <br> High: 1.10 ± 1.15 | Med: −1.92e-05 ± 1.41 <br> High: −2.20e00 ± 1.63 |
| Nervousness | 38.99 | 9.37 | 0.01 | Med: 1.52 ± 2.21 <br> High: 3.20 ± 2.53 | Med: −0.02 ± 0.06 <br> High: −0.11 ± 0.08 |
| Body condition | 39.46 | 9.84 | 0.01 | Med: 0.80 ± 0.66 <br> High: −0.03 ± 0.79 | Med: 0.03 ± 0.04 <br> High: 0.06 ± 0.05 |
| Brain volume change | 40.74 | 11.12 | 0.00 | Med: 0.08 ± 1.37 <br> High: 2.02 ± 1.45 | Med: 11.55 ± 23.66 <br> High: −4.28 ± 26.66 |
| Relative brain volume | 40.74 | 11.12 | 0.00 | Med: 0.72 ± 0.62 <br> High: −0.01 ± 0.73 | Med: −0.51 ± 1.23 <br> High: −1.10 ± 1.56 |

The model for age failed to converge properly because there were 0 instances of a subadult mastering the task. Intercept, coefficient, and SE estimates are given in logit scale.
†Binomial variable coefficients are for Subadult (Age) and Male (sex).
Most competitive model in bold.

adopted tool use to solve the task; these individuals were exclusively adults and primarily females.

Why are small adult females so proficient at solving the Aesop's task; conversely, why are adult males the least proficient? We posit that their smaller size imposes additional cognitive demands on the smaller females to compete with their male conspecifics – which can simply rely on their dominant status resulting from their size and strength – for access to resources. For example, when many crows gather around an ephemeral food source, dominant birds can immediately gain access to it by simply displacing smaller conspecifics, whereas the smaller crows are forced to assess the situation and choose an appropriate strategy; if there is enough food they can patiently wait until all the larger birds become sated and depart, if they have allies nearby they can attempt to recruit them to collectively displace their competitors[52], if the dominant birds become temporarily distracted they can dart in to grab food, if a satellite conspecific has succeeded in acquiring food from the main source they can attempt to steal it, if they know of another potential food source nearby they can leave in search of less competition, etc. The number of these potential strategies for subordinate birds suggests that cognitive flexibility and creativity might be advantageous in subordinate birds. Three of the four high-proficiency birds had relatively high body condition upon capture (Table 1 in SI), suggesting that they were consistently successful in acquiring food despite their smaller size relative to their peers. This would also explain the intermediate performance of the subadults: they're less dominant than the large adult males (thus, subjected to the same pressures as the adult females), but they lack the experience of the adult females. This hypothesis is further supported by other species with a female bias for tool use; female chimpanzees (*Pan troglodytes*), bonobos (*Pan paniscus*), and bottlenose dolphins (*Tursiops* sp.) are all smaller[53,54] and more likely to use tools[25,55] compared to their male conspecifics.

Although we hypothesized that the level of nervousness, body condition, and relative brain volume might also play a role in influencing a crow's eventual task proficiency, we did not observe any meaningful effect of those factors on tool proficiency. Our experimental design may have minimized any effect these factors may otherwise have had on tool proficiency. The habituation process and lengthy amount of time each crow was given to interact with the task during their training sessions may have allowed even the most nervous

crows ample opportunities to interact with the task. We calculated the crows' body condition using their weight upon capture, yet they had regular access to food in the days between their capture and their first imaging session, which meant the crows with low body condition values had likely improved their body condition by the time they began learning the task, thus lowering their expected motivation to acquire food. Finally, although brain volume has been found to correlate with the ability to solve cognitive tasks[56], other factors such as neuronal count, neuronal density, relative size of specific regions, or relative complexity of specific circuits within the brain might have been better measures of cognitive ability[10,57].

In conclusion, our use of American crows – an occasional tool user that struggled to learn true tool use during our study despite possessing the necessary mental ability and physical adaptations – has revealed crucial insights into the physical and neural attributes that contribute to regular tool use by a species. Unlike the more tool-adept New Caledonian crows[12], most American crows have little need to use tools in their environment[58]. However, smaller adult females have comparably more need than the larger males to creatively examine their environment for novel ways to access otherwise inaccessible food sources, which would account for their success at solving the Aesop's fable task presented in this study. Our findings support preexisting evidence[20] documenting that New Caledonian crows have enlarged mesopallium, striatal complex, septum, and tegmentum compared with non-tool-using species, and further indicate that these brain regions are all important, but differentially utilized across stages of tool use as birds learn, practice, and master new tools in their environment. Crows use their renowned intelligence to learn to use a tool initially, but they switch to circuits associated with motor learning and memory as they grow more familiar with it- a shift comparable to the changes in human brain activity after mastering a skill[46,47]. This finding could explain why so many vertebrates can master tool use independently after human trainers help them through the initial learning process (see[2-4] for examples). Broadening the application of such longitudinal studies of brain activity during the learning of difficult tasks could reveal common and unique aspects of neural networks across animal taxa; even though their common ancestors diverged more than 300 million years ago[59], birds and mammals show remarkably similar brain activity as they learn and master cognitively difficult tasks such as tool use.

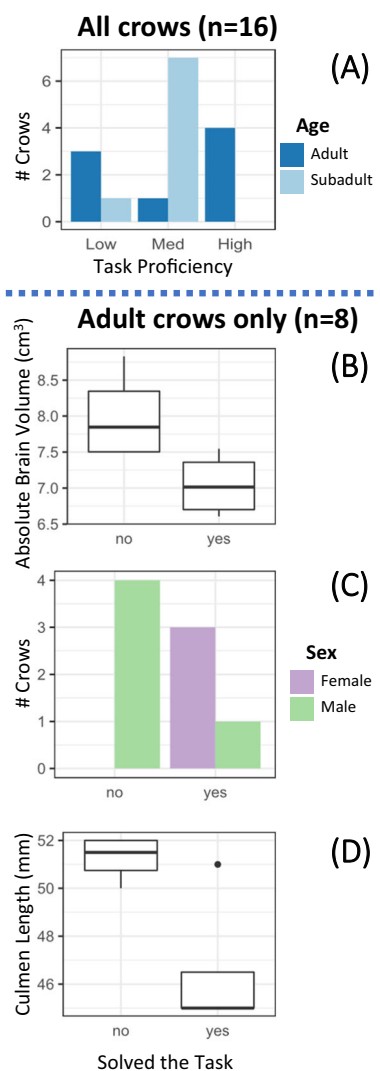

**Fig. 5 | Individual factors and task proficiency.** The number of adults and sub-adults (**A**) in each task proficiency group. Among the adult birds, absolute brain volume (**B**), sex (**C**), and size (represented by culmen length; **D**) were associated with a crow's ability to solve the task (achieved high-proficiency). **B** and **D** box-whisker plots display median (center line), upper/lower quartiles (box limits), and highest/lowest value within 150% of inter-quartile range (whiskers). Data beyond whiskers are displayed as outlier points. Source data are provided as a source data file.

## Methods

### Capturing and housing crows

We captured 16 wild American crows from various locations in Seattle, Bothell, and Woodinville WA as they departed their communal roost by luring them with bread and then trapping them using a net launcher (Fig. 1A). Due to potential differences in behavior and cognitive development, we immediately released birds in their first year of life. We caught two cohorts of birds outside of the breeding season and held each for several months ($n$ = 8 from October 2017 to February 2018, and $n$ = 8 from September 2018 to February 2019). We kept crows in a protected outdoor aviary at the University of Washington, Seattle. The crows were individually housed in adjacent cages (measuring 1.8 × 2.1 × 2.4 m; hereafter aviary cage) separated by wire mesh; crows could see and hear their neighbors but could not leave their cage. We provided crows with a rotating diet of assorted meats, eggs, grain, fruit, and dried dog kibble ad libitum for several hours each day.

### Aesop's fable task apparatus

We constructed the Aesop's task apparatuses using clear plastic acrylic tubes with an internal diameter of 5.1 cm mounted to the center of a (15.2 × 20.3 × 0.6 cm) polycarbonate sheet. We used cheese puffs as the food reward because they float on water for >3 h, possess a bright orange color that is easily seen, and are preferred by crows. We used plastic aquarium stones as the task's tools; they were relatively light (thus easy for a crow to lift), colored green or blue (increasing their contrast with the white arena), and, when dropped, each stone displaced water within the tube by 5–9 mm.

### Habituation/learning the Aesop's fable task

To reduce the neurological activity associated with the novelty of the procedure itself, we regularly (every 1–2 days) habituated crows to the unfamiliar conditions they would experience during the imaging process (Fig. 1B). All crows experienced >7 habituation sessions (mean ± SD: 10.2 ± 4.5 days) before their first scan.

We simulated the radiotracer injection by removing crows from their aviary cage, covering their heads with a cloth, laying them on their back, spraying their belly with water (to mimic the alcohol used as an antiseptic), and lightly pinching the injection site for several seconds. After the simulated radiotracer injection, we replicated the stimulus phase of the imaging process by moving the crow to a secluded area away from the main aviary and placing it inside a smaller cage (1 × 0.5 × 0.5 m; hereafter training cage), before moving a hollow wooden arena of identical size (hereafter the training arena) in front of the training cage. We placed the training arena directly adjacent to (and flush with) the front of the training cage and covered the cage's walls so that the crow's only external view was the interior of the arena. We painted the arena white and kept its interior well-lit by a ceiling-mounted LED light source (whereas the cage's interior was relatively dark due to the wall coverings) to further mimic the stimulus phase of imaging experiments. During the pre-scan habituation process, we kept the training arena's interior empty and prevented crows from leaving the training cage by keeping the cage door closed. We habituated crows before receiving their daily food to replicate the time after the last eating they would experience before the scan. After 2–3 h in the training cage, we returned the crows to their aviary cages and fed them.

After they were exposed to an Aesop's task during their first imaging session, we began the training phase (Fig. 1D). We continued to simulate the radiotracer injection before placing crows inside the training cage. However, we now allowed crows to enter the training arena and interact with an Aesop's fable apparatus. As before, we did not feed crows before training, ensuring they were motivated to retrieve the food reward. We returned crows to their aviary cage and fed them after 2–3 h or after they retrieved the food reward, whichever occurred first. We remotely tracked their progress using peephole viewers mounted to the training arena and recorded all training trials using a video camera placed inside the training arena.

We incrementally trained crows to solve the Aesop's fable task. We started by leaving 2–3 cheese puffs on the ground of the training arena to entice crows to enter the arena. After a crow began retrieving food from the arena floor, we added a baited Aesop's task apparatus with water filled to the brim and multiple stones along the tube's base. Each time a crow successfully retrieved food from the tube, we decreased the water level until we had established the maximum reachable depth for that crow. We then lowered the water depth to 5 mm below the crow's maximum reachable distance and attached training stones to lines emanating from inside the tube (Fig. 1iv). We balanced the training stones on the tube's rim so that as crows attempted to reach the food, they usually dropped one or more into the tube, bringing the water level and floating cheese puff up to a reachable distance. We next began increasing the length of the training stone lines, lowering the training stones along the tube exterior, forcing the crow to either lift

**Table 2 | Individual variable model selection (binomial) for an adult crow's likelihood of fully mastering the task**

| Model | AICc | Δ AICc | Wi | Intercept ± SE | Coefficient ± SE | P |
|---|---|---|---|---|---|---|
| Absolute brain volume | **11.22** | **0** | **0.29** | **76.32 ± 95.71** | **−10.22 ± 12.79** | **0.42** |
| Sex† | **11.4** | **0.18** | **0.27** | **19.57 ± 6208.83** | **−20.95 ± 6208.83** | **0.99** |
| Size | **11.5** | **0.28** | **0.25** | **46.97 ± 39.73** | **−0.95 ± 0.78** | **0.23** |
| Null | 13.76 | 2.54 | 0.08 | 7.85e-17 ± 7.07e-01 | | 1 |
| Body condition | 15.13 | 3.91 | 0.04 | −0.11 ± 0.83 | 0.10 ± 0.08 | 0.20 |
| Nervousness | 15.78 | 4.56 | 0.03 | 3.22 ± 2.83 | −0.11 ± 0.09 | 0.25 |
| Brain volume change | 16.84 | 5.62 | 0.02 | 1.11 ± 1.61 | −20.54 ± 26.19 | 0.43 |
| Relative brain volume | 17.11 | 5.89 | 0.02 | −0.03 ± 0.73 | −0.97 ± 1.62 | 0.55 |

The model for sex failed to converge properly because there were 0 instances of a female failing to master the task. Intercept, coefficient, and SE estimates are given in logit scale.
†Binomial variable coefficients are for Male (sex).
Most competitive models in bold.

the stones into the tube or pull them inside using the line before the water level was high enough for them to reach (Fig. 1v). Finally, we removed the training stones so that the crow was forced to pick stones off the ground and drop them into the tube before retrieving the food (Fig. 1vi); we considered any crow that did so to have mastered the task. If any crows failed to solve for 3 consecutive sessions, we reduced the difficulty by raising the water level or position of the training stones.

We continued to train each crow until they received their 2nd imaging session, which was logistically dictated by the availability of the PET scanner. The 2017 cohort of birds received a balanced number of training days (mean ± SD: 51.4 ± 0.5 days) because we were able to complete all birds' post-training scans within 15 days. However, we had limited access to the PET scanner for the 2018 cohort (45 days separating first and last post-training scans), which resulted in more variable training time for that group (53.8 ± 13.6 days). Because of this, we preferentially selected individuals that had stopped engaging with the apparatus for the earlier post-training scans to give additional training time to the birds that were still actively interacting. This variation in training time did not cause any significant differences in task proficiency between the two cohorts ($X^2$(2df, $N$ = 16) = 1.5, $P$ = 0.47).

We assigned crows into one of three proficiency categories for analysis. Low-proficiency crows never dropped any training stones into the tube, medium-proficiency crows occasionally dropped training stones while they were balanced on the tube lip (Fig. 1iv), and high-proficiency crows fully mastered the task by collecting stones from the ground and dropping them inside the tube. Variations within these categories can be found in Supplementary Tables 1 and 3 in the SI. The crows quickly progressed through the initial proficiency levels, with the low- and medium-proficiency birds reaching their maximum proficiency in 7.0 ± 3.7 days and 17 ± 10.5 days, respectively. However, it took much longer (48.0 ± 9.8 days) for the high-proficiency crows to begin deliberately dropping hanging training stones (Fig. 1v) into the apparatus, though they rapidly progressed from there to picking up stones from the ground and dropping them into the tube (3.0 ± 1.0 additional days). See Supplementary Fig. 9 in SI for each crow's learning curve.

Note that crows are social animals that are capable of social learning, thus teaching the crows to solve the task in isolation may not have been the most conducive way to train crows. However, previous research has shown that crows do not easily learn to solve tasks using social learning, although they will use it to refine their technique[60].

### Individual measures
To correlate individual physical characteristics with task proficiency, we measured and compared the following attributes from each bird: age, body condition, culmen length, level of nervousness, sex, brain volume (both absolute and relative to body size), and change in brain volume between the pre- and post-training scans. For age, we

categorized birds as being in their second year of life (subadult) or older (adult) using a combination of plumage color (dark brown/dull black for subadults, glossy black for adults), mouth coloration (traces of pink for subadults, fully black for adults), and feather wear (uniformly worn for subadults, mostly new for adults)[61]. We calculated the crows' body condition by extracting the residuals from a regression of their body weight upon capture against their culmen length (mm from the distal tip of the bill to the base of the feathers). We determined each bird's level of nervousness by standing 2 m away from each bird's cage while staring at a fixed point within the cage (not at the crow) and counting the bird's movements for 60 s. We assigned a numerical value based on the bird's perceived urgency to each move; walking along the perch = 0.5 or flying/hopping to another perch = 1. We obtained two such measurements for each bird (the first within two weeks of capture, the second within two weeks of release) and averaged them. We sexed our birds using a QIAGEN® DNeasy® Blood & Tissue Kit to isolate genomic DNA from each blood sample, amplifying the target genes (CHD1-W and CHD1-Z) using polymerase chain reaction (PCR), and conducting agarose gel electrophoresis on the PCR product to reveal sex differences[62].

We calculated brain volume using an open-source DICOM viewer Horos version 3.2.1 (The Horos Project, sponsored by Nimble Co LLC d/b/a Purview, Annapolis, MD, USA) to analyze the CT images obtained during the birds' first imaging session. We conducted brain segmentation on approximately 50% of the relevant slices using Horos's threshold-based 2D region of interest (ROI) utility, though we avoided the regions (cerebellum and brainstem) caudal to the nidopallium caudale; we edited these ROIs by hand before the software added interpolated ROIs on the missing slices. We inspected all generated ROI's before calculating the final brain volume ($cm^3$) with the built-in utility. Because some of the brain segmentation was performed manually (and thus subject to user bias), multiple researchers independently analyzed several of the same CTs; because our findings differed by a very small amount (0.82% ± 0.62%, $n$ = 11), we discounted user bias as minimal. To account for the allometric association of brain volume to body size, we extracted the residuals from a regression of brain volume and culmen length and used these residuals as a measure of relative brain volume during analysis in addition to raw absolute brain volume. Finally, we discovered that the crow's brain volumes were significantly smaller during their post-training scan (mean ± sd = 7.15 ± 0.63 $cm^3$) than they were during their pre-training scan (7.56 ± 0.68 $cm^3$, $t$(15) = 7.39, $p$ < 0.001), so we included the % reduction in volume between the two scans as a final individual measure against task proficiency (see Supplementary Table 1 in SI). An analysis is underway to investigate the reasons for the change in brain volume.

### Brain imaging
We imaged up to three crows per day, using a Siemens Inveon PET/CT system. We acquired usable imaging data from 14 of the 16 crows, with

the two individuals ($n = 1$ low-proficiency and $n = 1$ medium-proficiency) excluded because of mechanical/software issues with the imaging system during their scans. The scanning process consisted of a 20 min microPET scan, followed by a CT scan in the docked and coregistered microCT scanner. The scanners share a multimodality bed and have a bore diameter of ~12 cm. The PET field of view was approximately $8 \times 13$ cm$^2$ while the CT field of view was 7.9 cm $\times$ 13.3 cm; both included the entire brain with a slice thickness of approximately 0.1 mm. The scanner bed contained a pressure pad, which we used to monitor the crow's breathing (and thus the depth of anesthesia) during the scan process.

The evening before a bird was scanned (typically 1600–1700), we removed it from its aviary cage, placed it in a small animal carrier, and carried it across campus to the imaging laboratory, where we transferred it to a small wire cage with identical dimensions as the training cage ($1 \times 0.5 \times 0.5$ m; hereafter referred to as imaging cage) in a separate holding room to acclimate overnight. The imaging cage contained water but not food, ensuring crows fasted for at least 14 h before imaging to control for variable blood glucose levels influencing FDG uptake. The holding room's lights were set to follow a 12-hour day/night timer so the ambient light did not disrupt the crows' sleep cycle, and two sets of doors separated the holding room from the scanning room, ensuring the birds received minimal disturbance from the scanner or personnel.

The following morning, twenty minutes before the experiment, we covered the crow's imaging cage with a blanket (preventing it from looking out) and moved it into the scanning room to acclimate to the ambient noise of the room. To administer the radiotracer, we reached under the covering into the cage, grabbed the crow, covered its head with a cloth to calm it, placed it on its back, sprayed its belly with a disinfectant, and administered an intraperitoneal injection of approximately 1 mCi of [18 F] Fluorodeoxyglucose (FDG) (exact volume adjusted to account for radioactive decay and the bird's weight, ranging from 0.05 to 0.10 mL). After injection, we returned the crow to its covered cage and removed the cloth covering its head.

We presented the experimental stimuli within a wooden stage ($1.2 \times 0.6 \times 0.45$ m; hereafter the stimulus stage), the purpose of which was to block the crow's view of the imaging laboratory/personnel (thereby removing potential confounding sources of distraction) and to standardize the background color, light intensity, and light angle between trials. The stimulus stage was painted white to meet facility regulations for disinfecting porous surfaces and contained a ceiling-mounted LED light source directly above the stimulus. The front of the stimulus stage had two overlapping sliding panels, which we used to reveal or hide the stage interior; in addition to blocking the crow's view of the stimulus, the panels also blocked nearly all the light from the internal LED, increasing the contrast between showing and hiding the stimulus. The panels opened from the center of the crow's view of the stage so that the crow's eyes received equal stimulation, thus ensuring our methodology was not responsible for any bilateral differences in activation between the two brain hemispheres.

During the 3 min immediately after injecting the radiotracer, we positioned the stimulus stage in front of (and adjacent to) the covered imaging cage and removed the blanket from the cage side facing the stage; the crow remained in relative darkness because the remaining blanket blocked all exterior view of the lab while the stimulus stage's closed sliding panels prevented the crow from seeing the illuminated stage interior. At 3 min post-injection, we opened the sliding panels to reveal the experimental stimulus: a baited Aesop tube task with water level 12 cm from the top accompanied by seven stones. For the following 10 min (hereafter the stimulus phase), we used the sliding doors to alternatively reveal the stimulus to the crow for 60 s, then hide it for 30 s (seven exposures and six associated breaks total). After the final exposure ended at 13 min post-injection, we again removed the crow from the cage, covered its head with a cloth, and anesthetized it via a custom nose cone with 5% isoflurane in oxygen with a flow rate of 300–800 mL/min before placing it in the scanner (we reduced isoflurane concentration to 2.5–3% after the crow was fully induced).

We used Velcro straps to secure the anesthetized crow to the multimodality bed before starting the scanning process 26 min post-injection. The multimodality bed contained a pressure pad, which we used to monitor the crow's breathing (and thus the depth of anesthesia) during the scan process. After the scan was complete, we secured the crow in hand until it fully emerged from anesthesia (indicated when it regained the ability to grip with both feet), before returning it and the imaging cage to the holding room. We kept the scanned crows in the holding room for 20 h (the time required for 18 F radioactivity to decay to acceptable levels), after which we returned them to the aviary.

## Stimulus phase behavior

We observed the crows' behavior while they were metabolizing the FDG radiotracer and attending to the Aesop apparatus stimulus during the scanning process (see Supplementary Methods in the SI for details). The crows visually attended to the Aesop's fable task whenever it was revealed, gazing into the stimulus stage for much of the time (left eye: mean ± SD; 383.2 ± 37.5 s; right eye: 368.7 ± 49.6 s) that it was visible to them (420 s) without favoring one eye over the other ($t_{24} = 1.45$, $P = 0.16$) and with no significant difference in gaze time between the two scans (left eye: $t_{23} = 0.44$, $P = 0.66$; right eye: $t_{23} = 0.33$, $P = 0.74$). Their mean blink rate remained steady (23.38 ± 8.12 blinks/min) and did not change between scans ($t_{23} = 0.30$, $P = 0.77$). Although a single outlier individual from the low-proficiency group paced constantly and moved more than five times the amount of any other crow (398 movements), the remaining birds moved substantially less often during the stimulus phase (without outlier: 21.73 ± 22.78 movements, min = 0, max = 75). See SI for details on how we recorded the crows' behavior during the stimulus phase and Supplementary Fig. 8 in SI for behavioral changes between scans and proficiencies.

## Image processing

After we imaged each crow, we conducted a 13 min attenuation scan, then reconstructed the image using the vendor-supplied (i.e., Siemens Inveon) 3D OSEM/MAP algorithm with two 3D OSEM iterations followed by 18 MAP iterations and a target isotropic spatial resolution of 2.5 mm full width at half maximum, with attenuation and scatter corrections applied to the data. We imported the raw DICOM data to ImageJ[63], manually aligned their orientation to match an established jungle crow (*Corvus macrorhynchos*) brain atlas[64] that was adapted for PET[16], and trimmed the images to include only the brain. We stereotactically aligned the scans by estimating and applying nine affine parameters to the images using NEUROSTAT[65], an algorithm originally designed for automated human brain analysis which we adapted for crow brains analysis. The stereotactic alignment also corrected for differences in brain volume. We estimated alignment precision to be one-two pixels. We normalized all uptake values to a global brain FDG uptake. We used a jungle crow atlas[64] and a carrion crow (*Corvus corone*) atlas[66], adjusted by the differences in species brain size and the shape of whole versus extracted brains[67], as guides to identify the regions significantly activated by each stimulus.

## Statistical analysis

We conducted all statistical tests using R version 3.6.3[68]. We compared stimulus phase behaviors (blink rate, gaze, and movement) between scans using a Student's $t$-test and between proficiency groups using a linear model. We correlated FDG uptake with blink rate and movement using a Pearson correlation test. To determine which (if any) of the eight individual factors were associated with a crow's proficiency at solving the task, we constructed multinomial models using R package plyr[69] for every individual variable (along

with an accompanying null model) and used AICc to determine which model was most supported by the data. We considered models to be competitive if they were within 2 AICc of the model with the lowest AICc. Because the high-proficiency birds were exclusively adults, we conducted a second model selection among them with a binomial response "did/did not reach high-proficiency" using generalized linear models (family = binomial). Both models used a logit link. Because many variables were correlated with each other (such as sex and culmen length)[27], we report the Pearson's r between variables.

Because the crows' brains unexpectedly decreased in volume in the time interval between their pre- and post-training scans, we treated the two scans as independent for the voxelwise statistical comparison, despite having come from the same bird. We determined significant differences in regional activity using NEUROSTAT[65]. This algorithm conducts a Z-test comparing the study population's globally normalized difference in FDG uptake between the first (control) and second (stimulus) scans against the study population's pooled variance; it does this for each voxel coordinate throughout the entire brain using a modified Bonferroni correction with a smoothing factor. We only report the activity with a Z-score greater than 4.0 to reduce type I errors and to be consistent with previous imaging studies[16–19]. We also report changes in activity with a Z-score between 3.7–4.0 if the extent of activity is consistent with the size/location of a known brain region, as this may be worth examining in greater detail in future studies. As an additional verification of the voxelwise results, we ran an independent analysis, which sampled all the image sets from all subjects using spherical volumes of interest (VOIs; 2-voxel radius) centered around the significant peak coordinates. We plotted these VOI uptake values as a distribution to determine if said results are driven by outlying individual scans.

## Ethical Note

We captured, housed, and tested all crows (including PET/CT scans) in accordance with the Institutional Animal Care and Use Committee of the University of Washington (IACUC; protocol number 3077-01), Federal Collecting Permit MB761139-0, and State of Washington Scientific Collection Permit 14-010. All were released back into the wild after the study.

## Reporting summary

Further information on research design is available in the Nature Portfolio Reporting Summary linked to this article.

## Data availability

The source data for all tables and figures (i.e., VOI coordinates, normalized uptake values, crow individual measures, crow training progress, and crow behavior during FDG uptake) generated in this study have been deposited in the Dryad database and can be found here: https://doi.org/10.5061/dryad.83bk3j9xx. Acquired DICOM data from PET/CT are available on request. Source data are provided with this paper.

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

## Acknowledgements

We thank A. Lehnert and R. Miyaoka for their assistance with the imaging program, and the many undergraduate volunteers who assisted with animal care. We also thank E. Brenowitz, B. Gardner, and D. Perkel for their input on experimental design. J. DeLap created the crow silhouettes used in Fig. 1. This research was funded by the Eastern Band of Cherokee Indians Higher Education and Training Program (L.T.P), the NSF GRFP (L.T.P), the Seattle ARCS Foundation (L.T.P), the James Ridgeway Professorship (J.M.M), the NIH (grant: 1S10OD017980-01) (J.M.M), and the NERC (grant: NE/J018694/1) (C.N.T).

## Author contributions

L.T.P: Conceptualization, Funding acquisition, Animal care, Investigation, Data curation & analysis, Writing – original draft & revisions. J.M.M: Conceptualization, Investigation, Funding acquisition, Project administration, Supervision, Writing – review & editing. D.J.C: Methodology, Supervision, Data curation & analysis, Writing – review & editing. T.S: Methodology, Validation, Writing – review & editing. C.N.T: Methodology, Funding acquisition, Writing – review & editing.

## Competing interests

The authors declare no competing interests.
