## [Peer Review File · Nature Communications]

REVIEWER COMMENTS

Reviewer #1 (Remarks to the Author):

This is a very interesting study on the neural activity patterns that are associated with learning a cognitive task that involves tool use in American crows. It is definitely of general interest, the behavioral and overall technical parts are well conducted. Unfortunately, the identification and interpretation of the neuroanatomical findings show important weaknesses.

Anatomical interpretations:

Lines 78ff: The authors used the atlas of Izawa and Watanabe (2007) on jungle crows to identify areas of activity. A second option is the atlas of Kersten et al. (2022) on carrion crows. I used both to check the figures of the current study. Based on these two comparisons, it is highly likely that the lengthy strip of activity on the far left of figure 2 does not (or does mostly not) reflect a mesopallial but a nidopallial activity (please be aware that the AP-values of Kersten et al are slightly different from those of Izawa & Watanabe). This changes the functional interpretation since probably also in corvids this part of the nidopallium represents the trigeminal association area (Wild et al., 1984 *Brain Res.*, 208-215; Stacho et al., *Science*, 2020). So, associative trigeminal processes were very likely also active during pretraining.

Lines 215-222: The statement that "the nidopallium and its various sub-regions, such as the NCL, have been compared to the mammalian prefrontal cortex in controlling executive function ..." is wrong. Eight percent of the nidopallium even receive thalamopallial sensory input and most nidopallial areas process sensory associative information. Only the NCL is seen as a functional equivalent to PFC. This is important since the authors assign most caudal nidopallial activity as executive. A study by von Eugen et al (*J. Comp. Neurol.*, 2020) identified the location of NCL subareas in crows. These are shifted relative to the data from pigeon. The atlas of Kersten et al., (2022) incorporates them. The authors should locate their activity patterns accordingly.

Lines 224-227: It is a pity that the cerebellum is mentioned without any detail and only in passing. Since the cerebellum is highly topographically organized, its local activations could provide clues on the processes going on before and after tool learning. As far as I know, no detailed study on the crow cerebellum was ever conducted. But since this is a highly conserved area of the brain, the authors could use the pigeon and chick data to gain further insights (Wylie et al., *Front. Neurosci.*, 2018; Gutiérrez-Ibáñez et al., *J. Comp. Neurol.*, 2022; Schulte & Necker, *J. Comp. Physiol.*, 1998).

Lines 231-233: It doesn't make sense to report and discuss the finding within the tegmentum as "tegmentum", given that this is one of the most heterogeneous and complex areas of the brain. Looking in greater detail at the pictures, it seems as if the AVT was active, as a ventralmost tegmental area directly beneath the exist of the third nerve and directly caudal to the tuber. If this would be the case (and I would like the authors to check this carefully), it could imply that the reward pathway would be selectively activated in the successful birds when they saw again the apparatus. This would be an important finding.

Lines 267ff: I don't think we should discuss in length non-significant data. Given the small number of animals, the probability for false-positive results is high.

Literature:

Overall, and as also visible from above, I had the impression that literature search could be improved. I give two examples.

Line 19: Here, Emery & Clayton (2004) is cited, giving the impression that they said something about corvids and great apes possessing equivalent neuronal counts. First, Emery & Clayton could not know that data pointing in this direction was to be published more than a decade later and second, this argument is not completely true (Olkowicz et al., *PNAS*, 2016, Kverková et al., *PNAS*, 2022; Ströckens et al., *J. Comp. Neurol.*, 2022). In fact, corvids and great apes only have equivalent neuron counts when comparing the sum of mesopallial and nidopallial neurons with those in the PFC of great apes (Ströckens et al., *J. Comp. Neurol.*, 2022).

Line 39: Here, a review on the neural basis of tool use in humans is cited when saying that the avian NCL is associated with tool use. To my knowledge, there is up to now only indirect evidence for this claim by an anatomical study that involved New Caledonian crows (Mehlhorn et al., *Brain, Behav. Evol.*, 2010).

Reviewer #2 (Remarks to the Author):

I believe this is an extremely interesting paper, reporting important and well-conducted research. I have only minor requests for clarification or completeness of references.

Line 275: I am not sure I have understood this statement for there is plenty of evidence for a left hemisphere bias in the visual Wulst (e.g. for evidence at the single cell level: Costalunga et al (2021). Light-incubation effects on lateralisation of single unit responses in the visual Wulst of domestic chicks. *Brain Structure and Function*, Mar 30. doi: 10.1007/s00429-021-02259-y).

Line 283: I think the reference here for lateralization in the avian brain needs updating (the paper cited is quite specific, is not the first evidence and it is not a review, thus I suggest to be integrated with the addition of some more recent and general review, e.g. Rogers et al (2013) *Divided brains*. Cambridge Univ. Press).

Line 289 here again consider adding updated literature (e.g. Vallortigara, G., Rogers, L.J. (2020). A function for the bicameral mind. *Cortex*, 124: 274-285. doi: 10.1016/j.cortex.2019.11.018)

Line 295: the authors stressed the role of the right hemisphere for spatial tasks, consider however (not as alternative but complementary explanation) that there is extensive evidence for a role of the right hemisphere in response to novelty which may be important as well (e.g. MacNeilage et al. (2009). *Origins of the left and right brain*. *Scientific American*, 301: 60-67. Vallortigara, G., Versace, E. (2017). *Laterality at the Neural, Cognitive, and Behavioral Levels*. In "APA Handbook of Comparative Psychology: Vol. 1. Basic Concepts, Methods, Neural Substrate, and Behavior", J. Call (Editor-in-Chief), pp. 557-577, American Psychological Association, Washington DC).

RESPONSE TO REVIEWERS' COMMENTS

We thank the reviewers for their constructive comments and have revised our manuscript to reflect their suggested changes. We feel that the revised manuscript is greatly improved because of the reviewers' efforts. We detail our responses to each of their comments below and show those changes in the track-changes version of our revision. We also edited our writing for clarity and to meet the journal's formatting instructions.

Reviewer #1 (Remarks to the Author):

This is a very interesting study on the neural activity patterns that are associated with learning a cognitive task that involves tool use in American crows. It is definitely of general interest, the behavioral and overall technical parts are well conducted. Unfortunately, the identification and interpretation of the neuroanatomical findings show important weaknesses.

Anatomical interpretations:

Reviewer 1: Lines 78ff: The authors used the atlas of Izawa and Watanabe (2007) on jungle crows to identify areas of activity. A second option is the atlas of Kersten et al. (2022) on carrion crows. I used both to check the figures of the current study. Based on these two comparisons, it is highly likely that the lengthy strip of activity on the far left of figure 2 does not (or does mostly not) reflect a mesopallial but a nidopallial activity (please be aware that the AP-values of Kersten et al are slightly different from those of Izawa & Watanabe). This changes the functional interpretation since probably also in corvids this part of the nidopallium represents the trigeminal association area (Wild et al., 1984 Brain Res., 208-215; Stacho et al., Science, 2020). So, associative trigeminal processes were very likely also active during pretraining.

Author response: Thank you for bringing Kersten et al.'s (2020) atlas to our attention! We've incorporated Kersten et al.'s atlas as another source when determining which region which brain regions our significant loci correspond with. Because Kersten et al.'s and Izawa and Watanabe's (2007) atlases don't correspond exactly, we highlight any significant loci where the two atlases disagree over the active region (such as in the four regions in Figure 2 that are caudal to A9.4 being identified as N vs NC).

Based on the images it is difficult to say that the activity highlighted at A19.0 and A18.2 belongs to the nidopallium or mesopallium, as the atlases from both Izawa & Watanabe and Kersten et al. show a border between N and M at this location. Based on the overall shape of the super- and sub-threshold activation as it can be appreciated from approximately slice 15 to slice 27 (see Fig 8 in SI), we maintain our conclusion that this activity is more likely to be located primarily in the Mesopallium. In recognition of this uncertainty, we have updated our text to address the possibility of Nidopallial activity and that our interpretation is not conclusive.

Reviewer 1: Lines 215-222: The statement that "the nidopallium and its various sub-regions, such as the NCL, have been compared to the mammalian prefrontal cortex in controlling

executive function ...” is wrong. Eight percent of the nidopallium even receive thalamopallial sensory input and most nidopallial areas process sensory associative information. Only the NCL is seen as a functional equivalent to PFC. This is important since the authors assign most caudal nidopallial activity as executive. A study by von Eugén et al (J. Comp. Neurol., 2020) identified the location of NCL subareas in crows. These are shifted relative to the data from pigeon. The atlas of Kersten et al., (2022) incorporates them. The authors should locate their activity patterns accordingly.

Author response: Good point. We’ve rewritten some of our interpretation to acknowledge that most of our nidopallial activity was most likely associated with processing sensory information. Because the nidopallial activity was only present during the first scan, we’ve interpreted it as neophobic crows closely attending to the sights and sounds of the imaging process- they were novel during the first scan, less so during the second scan.

Reviewer 1: Lines 224-227: It is a pity that the cerebellum is mentioned without any detail and only in passing. Since the cerebellum is highly topographically organized, its local activations could provide clues on the processes going on before and after tool learning. As far as I know, no detailed study on the crow cerebellum was ever conducted. But since this is a highly conserved area of the brain, the authors could use the pigeon and chick data to gain further insights (Wylie et al., Front. Neurosci, 2018; Gutiérrez-Ibáñez et al., J. Comp. Neurol., 2022; Schulte & Necker, J. Comp. Physiol., 1998).

Author response: The part showing a high activity (Fig 4) is the caudal portion of the cerebellum, suggesting the activity might be related to the oculomotor cerebellum. We have included this information in the manuscript. However, with limited studies on the crow cerebellum, we hesitate to make further interpretations for this region.

Reviewer 1: Lines 231-233: It doesn’t make sense to report and discuss the finding within the tegmentum as “tegmentum”, given that this is one of the most heterogeneous and complex areas of the brain. Looking in greater detail at the pictures, it seems as if the AVT was active, as a ventralmost tegmental area directly beneath the exist of the third nerve and directly caudal to the tuber. If this would be the case (and I would like the authors to check this carefully), it could imply that the reward pathway would be selectively activated in the successful birds when they saw again the apparatus. This would be an important finding.

Author response: This is an interesting point that we hadn’t considered. We compared our observed activity with the Kersten et al. (2020) atlas, and it does roughly correspond with the VTA. However, as the tegmentum is heterogeneous and complex (and the VTA is a small region), we don’t want to outright declare this activity to be the VTA. We’ve decided to further describe the location of the activity in the Results and include the possibility of it being the VTA (along with the ramifications) in the Discussion.

Reviewer 1: Lines 267ff: I don't think we should discuss in length non-significant data. Given the small number of animals, the probability for false-positive results is high.

Author response: We have removed this paragraph.

Literature:

Overall, and as also visible from above, I had the impression that literature search could be improved. I give two examples.

Reviewer 1: Line 19: Here, Emery & Clayton (2004) is cited, giving the impression that they said something about corvids and great apes possessing equivalent neuronal counts. First, Emery & Clayton could not know that data pointing in this direction was to be published more than a decade later and second, this argument is not completely true (Olkowicz et al., PNAS, 2016, Kverková et al., PNAS, 2022; Ströckens et al., J. Comp. Neurol., 2022). In fact, corvids and great apes only have equivalent neuron counts when comparing the sum of mesopallial and nidopallial neurons with those in the PFC of great apes (Ströckens et al., J. Comp. Neurol., 2022).

Author response: This was the result of our citation tool not correctly updating the citation numbers after we removed a reference from an earlier draft. Emery & Clayton (2004) were meant to be cited for the statement “[Corvids and Apes] can solve novel challenges with similar speed and flexibility”. We have corrected this. We have also modified the neuronal count statement to reflect your feedback.

Reviewer 1: Line 39: Here, a review on the neural basis of tool use in humans is cited when saying that the avian NCL is associated with tool use. To my knowledge, there is up to now only indirect evidence for this claim by an anatomical study that involved New Caledonian crows (Mehlhorn et al., Brain, Behav. Evol., 2010).

Author response: This was another result of our citation tool not correctly updating the citation numbers after we removed a source from an earlier draft. The intended citation was Boire, D., Nicolakakis, N. & Lefebvre, L. Tools and brains in birds. *Behaviour* 139, 939–973 (2002). We will double-check that it is working as intended prior to submitting our revisions.

Reviewer #2 (Remarks to the Author):

I believe this is an extremely interesting paper, reporting important and well-conducted research. I have only minor requests for clarification or completeness of references.

Reviewer 2: Line 275: I am not sure I have understood this statement for there is plenty of

evidence for a left hemisphere bias in the visual Wulst (e.g. for evidence at the single cell level: Costalunga et al (2021). Light-incubation effects on lateralisation of single unit responses in the visual Wulst of domestic chicks. *Brain Structure and Function*, Mar 30. doi: 10.1007/s00429-021-02259-y.

Author response: We were specifically referring to evidence for left-hemisphere bias in the mesopallium ventro-lateralis, rather than the hyperpallium apicale and visual Wulst. However, we have removed this paragraph entirely per the other reviewer's feedback as all the regions covered in this section did not exceed our threshold for significance.

Reviewer 2: Line 283: I think the reference here for lateralization in the avian brain needs updating (the paper cited is quite specific, is not the first evidence and it is not a review, thus I suggest to be integrated with the addition of some more recent and general review, e.g. Rogers et al (2013) *Divided brains*. Cambridge Univ. Press).

Author response: Thanks for providing a more appropriate paper- we have replaced the original citation with the one you suggested.

Reviewer 2: Line 289 here again consider adding updated literature (e.g. Vallortigara, G., Rogers, L.J. (2020). A function for the bicameral mind. *Cortex*, 124: 274-285. doi: 10.1016/j.cortex.2019.11.018)

Author response: We have added your suggested citation here.

Reviewer 2: Line 295: the authors stressed the role of the right hemisphere for spatial tasks, consider however (not as an alternative but complementary explanation) there is extensive evidence for a role of the right hemisphere in response to novelty which may be important as well (e.g. MacNeilage et al. (2009). Origins of the left and right brain. *Scientific American*, 301: 60-67. Vallortigara, G., Versace, E. (2017). Laterality at the Neural, Cognitive, and Behavioral Levels. In "APA Handbook of Comparative Psychology: Vol. 1. Basic Concepts, Methods, Neural Substrate, and Behavior", J. Call (Editor-in-Chief), pp. 557-577, American Psychological Association, Washington DC).

Author response: Interesting! We were unaware of the link between the right hemisphere and novelty. We've updated this section to include this information.

REVIEWERS' COMMENTS

Reviewer #1 (Remarks to the Author):

The authors have properly responded to my comments and I'm now happy with the manuscript.